# Manipulating Attention Does Not Deceive Us

**Howard Caulfield, Artemis Capari, Willemijn de Hoop, Sander Kohnstamm**
University of Amsterdam
Amsterdam, Netherlands
{howard.caulfield, artemis.capari}@student.uva.nl,
{willemijn.dehoop, sanderkohnstamm}@gmail.com

*Template and style guide to ML Reproducibility Challenge 2020. The following section of Reproducibility Summary is **mandatory**. This summary **must fit** in the first page, no exception will be allowed. When submitting your report in OpenReview, copy the entire summary and paste it in the abstract input field, where the sections must be separated with a blank line.*

**Scope of Reproducibility**

The authors of our paper claim that attention weights can easily be manipulated without significant accuracy loss and that human subjects can be deceived by these attention weights. We will attempt to reproduce the former.

**Methodology**

We used their code which was publicly available on github. Their data was also included. We also utilised a cluster computer for its GPU performance. This was provided by the University of Amsterdam.

**Results**

Our results do reproduce the original results fairly well. There are some minor divergences, but nothing too significant that it would not uphold the authors claims. We have been able to reproduce 90% of the results within the error margins produced by differently seeded runs.

**What was easy**

The experience of the reproduction was relatively smooth overall. With very minor changes almost all code could run. The code was wel documented and well structured.

**What was difficult**

There was some missing code concerning the BERT model and masking functions. These posed a problem. Also, some data that the authors used was private. This prevented us of reproducing that part.

**Communication with original authors**

The communication with the authors was quick and to the point. They were able to help us with some of the missing code.

## 1 Introduction

The ability of attention explaining a model's decisions has become a highly debated topic ever since its inception [2]. An attention mechanism is an aggregation of the input tokens combined with weights. These weights allow the mechanism to theoretically change the importance of, or its *attention* over, different parts of the input at each output step.

As the weights seem to provide a clear and understandable overview of the *attention* a model gives to certain parts of the input, it has become a widely used tool for auditing algorithms in the context of fairness and accountability. However, there are many doubts about the reliability of this method as well. Ergo, many papers have been published in a relatively short period highlighting both sides of this problem. These papers often provide results that either prove a correlation between the attention weights and predictions [11][10], or the ambiguity between attention weights and the performance of the rest of the model [6][7], or somewhere in between [9].

The paper Learning to Deceive with Attention-Based Explanations [8] aims to prove that attention weights do not convey true interpretability. The authors of the paper make two main claims in their study:

- A model's attention weights can be manipulated without affecting its performance
- A human subject can be deceived by manipulated attention based explanation

The paper aims to support the first statement by performing a range of different models for classification and sequence-to-sequence tasks. It decreases attention weight on impermissible tokens, feigning no or little explanation of a prediction by these tokens, whilst maintaining high levels of accuracy. The attention reduction is achieved by a modified loss function that penalises the attention mass on these tokens. The second claim is supported by a qualitative human study, where three human subjects were asked questions about the explanation of a BiLSTM occupation prediction model. As the attention mass on the most biased tokens has been decimated, the human audience is easily deceived. Hence, the authors conclude that the model is learning to deceive. For our reproducibility study [1] we have chosen not to focus on the human study because we feel it lacks scientific basis and current circumstances complicate conducting human studies. Instead, we have aimed our attention at the classification and sequence-to-sequence tasks and their implications on interpretability of attention. One of the original classification tasks (the recommendation letter classification) we were not able to reproduce, since the data set is not publicly available. For all other models we summarised our reproduced findings and compared them to the original results.

## 2 Methodology

Pruthi et al. [8] introduce a method for manipulating attention that penalizes the attention weights of impermissible tokens. For an input sequence $S = w_1, w_2, ..., w_n$ each token has attention $\alpha \in [0, 1]^n$ such that $\sum_i \alpha_i = 1$. We define attention by the dot-product function: $A(Q, K, V) = (QK^T)V$, where $\mathbf{Q}$ is the query matrix, $\mathbf{K}$ the key matrix, and $\mathbf{V}$ the value matrix. By the dot product, the similarity between query and key is computed, higher similarity leads to higher attention. The softmax normalizes the attention scores. Depending on the task, a set of impermissible tokens is determined a priori and an $n$-dimensional binary vector $\mathbf{m}$ can be defined such that:

$$\mathbf{m} = \left\{ \begin{array}{ll} 1 & \text{if } w_i \in I \\ 0 & \text{otherwise} \end{array} \right.$$

Pruthi et al. [8] add a penalty $\mathcal{R}$ to the loss function that is used for the specific task, resulting in a total cost of $L' = L + \mathcal{R}$. The penalty for a single attention vector is is defined as follows:

$$\mathcal{R} = -\lambda \log \left( 1 - \alpha^T \mathbf{m} \right) \tag{1}$$

Where $\lambda$ is a coefficient that determines the magnitude of the penalty. The coefficient is therefore negatively correlated with the assigned attention mass of impermissible tokens. The penalty defined in equation 1 is suitable for simple neural networks that only have one attention layer. More extensive neural networks that incorporate multiple attention layers require a different definition of penalty $\mathcal{R}$ [10]. Pruthi et al. use two options for defining the penalty of multi-head attention:

---

[1]This work is produced as part of a course Fairness, Accountability, Confidentiality and Transparency in AI at the University of Amsterdam.

$$\mathcal{R} = -\frac{\lambda}{|\mathcal{H}|} \sum_{h \in \mathcal{H}} \log\left(1 - \alpha_h^T \mathbf{m}\right) \quad (2)$$

$$\mathcal{R} = -\lambda \min_{h \in \mathcal{H}} \log\left(1 - \alpha_h^T \mathbf{m}\right) \quad (3)$$

Equation 2 penalizes the mean value of attention weights of impermissible tokens, whereas 3 penalizes the maximum attention value from all heads.

For performing several classification tasks, Pruthi et al. [8] use three types of models. The first one is a simple embedding model, where word embeddings are aggregated following a weighted sum, the attention is equal to these weights. A linear layer and a softmax function determine the prediction of the input. The second model is a Bidirectional LSTM model with attention[5]. Both the embedding and BiLSTM model penalize attention following equation 1. The third model is Bidirectional Encoder Representations from Transformers (BERT)[4]. BERT models are able to pretrain deep bidirectional text representations and are therefore suitable for many NLP tasks. BERT is effective because of its recurrent information flow, but this complicates the task of masking the attention of impermissible tokens. We need to extend the attention mask as defined in equation 0 such that information flow between the impermissible tokens and the others is blocked. For an $n$ input tokens, we define the attention mask to be a matrix $\mathbf{M}_{ij}$ of size $n \times n$. If the ith and jth of an input sequence are in the same set ($I$ or $I^c$), $\mathbf{M}_{ij}$ is 1, otherwise it is 0. This type of self-attention is called multi-head attention, and there are two ways in which penalty can be calculated (equation 2 and 3). For the BERT model, we will manipulate both maximum and mean attention. This was adapted to BERT by placing masks on all attention layers such that only impermissible/permissible tokens could attend to impermissible/permissible tokens.The special token [CLS] (i.e. the token for the start of the sentence, could attend to any token but no other token could attend to it. This mask is used to ensure there is no information flow leaking from impermissible to permissible from the layer to layer.

For the sequence-to-sequence tasks only one model is used. This is an encoder-decoder model. The encoder part is a bidirectional Gated recurrent unit (GRU)[3]. The decoder is a unidirectional GRU. A Gated recurrent unit is very similar to a LSTM, but without an output gate. This reduces its number of parameters significantly. There is no further reason discussed for this model choice in the paper.

## 3 Experiments

We reproduce four binary text classification tasks and four sequence to sequence tasks as proposed by Pruthi et al. [8] . Additionally, we propose an extension of their paper by performing multi class sentiment analysis. For each task, we apply the models described in section 2. The goal is to analyze the extent of manipulability of attention and see how a model's performance changes when decreasing the attention scores. Every task has a corresponding set of impermissible tokens that is determined a priori of which the attention scores are aimed to be diminished. We run our models for different values of the penalty coefficient $\lambda \in \{0, 0.1, 1\}$, where $\lambda = 0.0$ shows a vanilla model without regulation of attention mass, and $\lambda = 1.0$ shows maximum regulation of attention. We use accuracy as a measure for the performance of each model. Moreover, we calculate the attention mass of the impermissible tokens. We measure the accuracy and attention mass of each task, model, and $\lambda$ over five seeds (unless stated otherwise), of which we will report its mean and standard deviation. Due to the large computational needs of the tasks, all models are run on a LISA cluster computer. The Lisa system is a collaboration of the University of Amsterdam (UvA), the Vrije Universiteit and the SURF organisation. This cluster computer allowed us to utilise various GPU's, including a 1080Ti, Titan V or a Titan RTX. We were limited to two GPU's at a time.

### 3.1 Reproduction

#### 3.1.1 Classification

Each classification task uses its own data set and is performed by four models to compare different types of attention manipulation for binary classification. The data sets are divided into a train-, development-, and test set, which are available for download[2]. The size of the data sets and their corresponding subsets are presented in Table 1. Additional to different values of penaly coefficient $\lambda$, we perform the different tasks with a complete anonymization or deletion of impermissible tokens. This serves as a ground check to compare the accuracy of manipulated attention weights to.

---

[2]https://github.com/danishpruthi/deceptive-attention/tree/master/src/classification_tasks/data

|              | Train   | Development | Test   | Total   |
| ------------ | ------- | ----------- | ------ | ------- |
| **Occupation** | 17.629  | 2.519       | 5.037  | 25.185  |
| **Pronoun**    | 9.017   | 1.127       | 1.127  | 11.271  |
| **SST+Wiki**   | 6.920   | 827         | 1.821  | 9.568   |

Table 1: Number of examples for each data set

**Occupation classification** is performed on biographies collected by De-Arteaga et al. [1] containing descriptions of two professions: surgeons and physicians. This task aims to predict which occupation a biography describes. More than 80% of the surgeons in this data set is male, making it suitable for studying gender bias in text classification. To make sure the models use gender indicators for prediction, the difference between female and male surgeons is enlarged. Due to the large discrepancy in gender for both occupations, we know the model relies on gender tokens. Therefore we consider pronouns and gendered name prefixes ("Mr", "Ms, etc.) as impermissible tokens. We anonymize impermissible tokens by substituting gender pronouns like "he" and "herself" by neutral pronouns such as "they" and "themself". Name prefixes like "Mr." and "Ms." will be changed to a gender neutral version "Mx", analogous to the neutral version of "Latino/a": "Latinx".

**Pronoun-based Pronoun** is performed on a data set containing biographies from Wikipedia where the subject of each biography is labeled as either male or female. Ambiguous biographies that do not contain any pronouns or contain pronouns from both genders[3] were removed from the data set. This implies that gender classification can reach an accuracy of 100% when looking at pronouns. We consider gender pronouns as impermissible tokens. Corresponding to the occupation classification task, we anonymize the impermissible tokens by neutralizing gender pronouns ("he" becomes "they").

**Sentiment Analysis with Distractor Sentences** combines two data sets, the Stanford Sentiment Treebank (SST) and Wikipedia pages. The goal is to identify the sentiment of a movie review with a randomly-selected distractor sentence from Wikipedia appended to it, separated by a token '[SEP]'. All words from the movie review are considered impermissible tokens. For the anonymization of the impermissible tokens, all SST reviews are simply deleted from the data set. This implies that the classification models with anonymized impermissible cannot outperform random guessing.

All models train over a set number of epochs. After each epoch, the development set is used to determine which model attains the largest reduction in attention mass while maintaining an accuracy that is within a 2% margin of the original accuracy. This model is then chosen for further training. The embedding model and the BiLSTM model have an embedding dimension of 128, the hidden size of the BiLSTM is 64. Both embeddings were trained from scratch. For the transformer model, we have imported Google's basic pretrained BERT model [4] that contains 12 layers with self attention. Two BERT models are further trained for every task, one that uses average attention as a metric for training (equation 2), and one that uses maximum attention (equation 3). A detailed overview of the hyperparameters per model and task is given in table **??**. We have not performed hyperparameter tuning, but mostly used hyperparameters from the original research. The classification tasks took a total of around 80 hours; with each model taking on average 20 minutes to train on one seed for a total of 240 seeds.

| Model | Dataset | Pretrained | $\epsilon$ | # Epochs | Hidden | # Batches | | Dimension Size | | |
| ----- | ------- | ---------- | ---------- | -------- | ------ | --------- | ---- | --- | --- | --- |
|       |         |            |            |          |        | Train | Eval | Emb | Hid | Out |
| Emb + Att   | Occupation | No  | $1e-12$ | 5  | -  | 1 | 1 | 128 | -  | 2 |
| Emb + Att   | Gender     | No  | $1e-12$ | 15 | -  | 1 | 1 | 128 | -  | 2 |
| Emb + Att   | SST+Wiki   | No  | $1e-12$ | 15 | -  | 1 | 1 | 128 | -  | 2 |
| BiLSTM + Att | Occupation | No  | $1e-12$ | 5  | 1  | 1 | 1 | 128 | 64 | 2 |
| BiLSTM + Att | Gender     | No  | $1e-12$ | 15 | 1  | 1 | 1 | 128 | 64 | 2 |
| BiLSTM + Att | SST+Wiki   | No  | $1e-12$ | 15 | 1  | 1 | 1 | 128 | 64 | 2 |
| BERT | Occupation | Yes | $2e-5$ | 4 | 12 | 32 | 8 | | | |
| BERT | Gender     | Yes | $2e-5$ | 4 | 12 | 32 | 8 | | | |
| BERT | SST+Wiki   | Yes | $2e-5$ | 4 | 12 | 32 | 8 | | | |

Table 2: Hyper Parameters

---

[3]We recognize that in society gender is a spectrum but for simplicity and scientific purpose we consider only two genders

[4]https://github.com/google-research/bert

### 3.1.2 Sequence-to-Sequence tasks

The sequence to sequence tasks were implemented to further improve the foundation of the authors claims. Not only were attention weights originally introduced for sequence to sequence tasks, they can also provide very clear impermissaile tokens in the form of golden alignments in certain data set settings. These tasks were all performed with an encoder-decoder architecture consisting of two GRU's.

These sequence to sequence tasks provide a unique opportunity to test tasks with *golden alignments*. The authors chose to implement three different tasks in which the order of the tokens is adjusted and one machine translation task. Each of the former tasks has a direct mapping of a certain input token to a certain output token. This is why these input-output token pairs can be seen as golden alignment, or, in case of the input tokens, very robust impermissible tokens.

As for a more realistic problem, the authors also have evaluated the performance on a machine translation task. They have used English to German translation from the Multi30K dataset. This dataset is comprised of image descriptions. As the golden alignment of each word is less clear, they also implement the Fast Align toolkit to provide them with these golden alignments.

As was mentioned above, the goal of the sequence to sequence tasks is to gain a better understanding of the flow of information after the manipulation of the attention weights, as well as prove that this method also works in the case that attention mechanisms were originally intended for. To this end the authors implemented the following tasks for the model to perform:

- **Bigram flipping**; here the goal is to reverse every bigram in the input.
- **Sequence Copying**; here the goal is simply copy the sequence.
- **Sequence Reversal**; here the goal is to reverse the entire input sequence.

As well as these simple tasks, the authors attempted to reproduce a more real world scenario by including a machine translation task.

for the sequence to sequence tasks around 50 hours; with each model taking on average half an hour to train on one seed for a total of 100 seeds.

For each of these tasks the experiment setup was fairly well put together by the authors. The code was ported to the LISA cluster computer and ran by installing their environment and running the bash files. The only minor adjustment that was needed was to comment out the import of the `log` package and all of its uses in the code, as was similar to all implementations of the paper.

The models are run for 30 epochs, or conversion before that. An average over 5 different seeds is taken as result. As for the loss function for the training of the three simple tasks an accuracy score is taken. For the machine translation task the BLUE score is utilised. There is no mention of further hyperparameter search in this paper.

## 4 Results

### 4.1 Reproduction

Our results for the reproduction of the classification and sequence-to-sequence tasks can be found in Table 3 and **??** respectively. In Table 4 the differences in accuracy and attention mass are portrayed. In general, the reproduced results seem to agree with the results from the paper. Some minor fluctuations occur but these are mostly in line with the margin of error of the different seeds. For the classification tasks, our measured accuracies are typically higher.

The accuracy of the models in occupation prediction and gender identification do not or barely decrease for manipulated attention weights. Some accuracies even increase slightly. In contrary, removing the impermissible tokens completely by anonymization shows a significant negative effect on the accuracy. Especially gender identification by the simple embedding and BiLSTM models yields very low accuracy scores (around 70%). The more complex BERT models seem to be able to handle the anonymization better and keep an accuracy of 80%. For the sentiment analysis task, the performance of the embedding and BiLSTM model decreases along with the attention scores. The transformer models' accuracy remains constant for different values of $\lambda$

The reproduced result of the sequence to sequence tasks are given in table 5. These show very minimal offset from the original results. Only uniform the uniform attention mass seems to indicate a general 10% dip in performance over all attention weight variations and the BLUE accuracy score of the machine translation has around a 10% increase over all models.

| Model | $\lambda$ | $\mathcal{I}$ | Occupation | | Gender | | SST+Wiki | |
|---|---|---|---|---|---|---|---|---|
| | | | Acc. | A.M. | Acc. | A.M. | Acc. | A.M. |
| Embedding | 0.0 | ✗ | 93.26 ± 0.27 | - | 70.81 ± 1.73 | - | 49.35 ± 1.15 | - |
| Embedding | 0.0 | ✓ | 96.52 ± 0.18 | 54.43 ± 1.88 | 100.0 ± 0.0 | 97.77 ± 1.99 | 71.66 ± 0.95 | 51.59 ± 1.05 |
| Embedding | 0.1 | ✓ | 96.26 ± 0.23 | 3.68 ± 0.56 | 100.0 ± 0.0 | 4.06 ± 1.41 | 70.12 ± 1.1 | 17.27 ± 1.16 |
| Embedding | 1.0 | ✓ | 96.19 ± 0.11 | 0.9 ± 0.29 | 99.84 ± 0.13 | 0.84 ± 0.06 | 51.04 ± 0.67 | 11.54 ± 2.37 |
| BiLSTM | 0.0 | ✗ | 93.76 ± 0.08 | - | 70.74 ± 0.58 | - | 49.75 ± 0.39 | - |
| BiLSTM | 0.0 | ✓ | 96.67 ± 0.21 | 46.43 ± 6.39 | 100.0 ± 0.0 | 95.71 ± 3.89 | 77.55 ± 0.96 | 82.9 ± 1.85 |
| BiLSTM | 0.1 | ✓ | 96.52 ± 0.05 | 0.04 ± 0.01 | 99.98 ± 0.04 | 1.78e-06 ± | 64.86 ± 2.04 | 0.35 ± 0.24 |
| BiLSTM | 1.0 | ✓ | 96.46 ± 0.2 | 2.83e-03 ± | 99.93 ± 0.14 | 1.80e-09 ± | 65.06 ± 2.53 | 0.05 ± 0.06 |
| BERT (mean) | 0.0 | ✗ | 95.51 ± 0.14 | - | 82.25 ± 0.14 | - | 50.32 ± 1.36 | - |
| BERT (mean) | 0.0 | ✓ | 97.23 ± 0.02 | 5.95 ± 4.06 | 99.86 ± 0.15 | 68.77 ± 16.77 | 91.76 ± 0.25 | 16.76 ± 5.16 |
| BERT (mean) | 0.1 | ✓ | 97.27 ± 0.17 | 0.01 ± | 99.93 ± 0.04 | 0.01 ± | 91.64 ± 0.14 | 0.05 ± 0.01 |
| BERT (mean) | 1.0 | ✓ | 97.3 ± 0.08 | 7.01e-04 ± | 99.88 ± 0.04 | 2.61e-04 ± | 91.75 ± 0.24 | 0.01 ± |
| BERT (max) | 0.0 | ✗ | 95.56 ± 0.12 | - | 82.25 ± 0.14 | - | 50.32 ± 1.36 | - |
| BERT (max) | 0.0 | ✓ | 97.23 ± 0.02 | 58.02 ± 40.05 | 99.86 ± 0.15 | 99.76 ± 0.33 | 91.76 ± 0.25 | 64.13 ± 11.45 |
| BERT (max) | 0.1 | ✓ | 97.05 ± 0.21 | 0.01 ± | 99.88 ± 0.04 | 4.76e-03 ± | 91.86 ± 0.04 | 0.06 ± 0.01 |
| BERT (max) | 1.0 | ✓ | 97.25 ± 0.11 | 7.86e-04 ± | 99.86 ± 0.07 | 5.28e-04 ± | 92.07 ± 0.2 | 3.88e-03 ± |

Table 3: Reproduction Results of the Classification Tasks

| Model | $\lambda$ | $\mathcal{I}$ | Occupation | | Gender | | SST+Wiki | |
|---|---|---|---|---|---|---|---|---|
| | | | Acc. | A.M. | Acc. | A.M. | Acc. | A.M. |
| Embedding | 0.0 | ✗ | 0.54 | - | -4.01 | - | -0.45 | - |
| Embedding | 0.0 | ✓ | -3.22 | -3.03 | 0.00 | 1.43 | -0.96 | -3.19 |
| Embedding | 0.1 | ✓ | -0.06 | 0.92 | -0.60 | -0.66 | -2.22 | 19.13 |
| Embedding | 1.0 | ✓ | 0.01 | 0.4 | -0.64 | -0.04 | -2.64 | -2.84 |
| BiLSTM | 0.0 | ✗ | -0.46 | - | -7.44 | - | -0.65 | - |
| BiLSTM | 0.0 | ✓ | -0.27 | 3.87 | 0.00 | 1.09 | -0.65 | -5.2 |
| BiLSTM | 0.1 | ✓ | -0.12 | 0.04 | 0.02 | 0 | -4.26 | -0.31 |
| BiLSTM | 1.0 | ✓ | 0.24 | 0.01 | 0.07 | 0 | -4.06 | 0.02 |
| BERT (mean) | 0.0 | ✗ | -0.51 | - | -9.45 | - | 0.08 | - |
| BERT (mean) | 0.0 | ✓ | -0.03 | 7.95 | 0.14 | 12.03 | -0.96 | 42.24 |
| BERT (mean) | 0.1 | ✓ | -0.07 | 0 | -0.03 | 0 | -0.74 | 0.05 |
| BERT (mean) | 1.0 | ✓ | -0.10 | 0.01 | 0.02 | 0.01 | -1.15 | 0 |
| BERT (max) | 0.0 | ✗ | -0.56 | - | -9.45 | - | 0.08 | - |
| BERT (max) | 0.0 | ✓ | -0.03 | 41.68 | 0.14 | -0.06 | -0.96 | 32.07 |
| BERT (max) | 0.1 | ✓ | 0.05 | 0 | 0.02 | 0.01 | -1.16 | 0.04 |
| BERT (max) | 1.0 | ✓ | 0.15 | 0.01 | -0.06 | 0 | -1.87 | 0.01 |

Table 4: Differences between our results and the results from Table 3 in Pruthi et al. [8]

# 5 Discussion

## 5.1 Reproduction

Our results are in line with the results produced by Pruthi et al. [8]. Therefore we can draw similar conclusions and state that a model's attention weights *can* be manipulated without affecting its performance.

The accuracy of occupation classification only decreases by 3% when anonymizing the impermissible tokens. The Gender identification accuracy decreases by 20% but is still a lot better than random guessing. These two results imply that there are words that could serve as proxies for the impermissible tokens. Not only pronouns and Mr., Ms. contain information about gender, but other words could also contain info.

- Some words can be proxy for gender words in occupation classification and gender identification

The tasks of gender identification and sentiment analysis both rely significantly on the set of impermissible tokens for high performance. This is obvious from the measured accuracy when the impermissible tokens are anonymized or deleted. When the attention scores on these impermissible tokens are decreased, accuracy for all models (except the embedding model) remain relatively high. This shows information from the impermissible tokens is spread across the

| Attention | | Bigram Flip | | Sequence Copy | | Sequence Reverse | | En → De MT | |
|---|---|---|---|---|---|---|---|---|---|
| | | Acc. | A.M. | Acc. | A.M. | Acc. | A.M. | BLUE | A.M. |
| Dot-Product | 0.0 | 100.0±0.0 | 93.89±0.16 | 100.0±0.0 | 94.11±0.12 | 99.99±0.01 | 93.49±0.95 | 37.58±0.72 | 24.85±1.82 |
| Uniform | 0.0 | 92.49±2.52 | 4.71±0.0 | 81.76±0.77 | 4.73±0.0 | 80.91±4.16 | 4.74±0.0 | 32.35±0.91 | 5.96±0.0 |
| None | 0.0 | 94.89±1.39 | 0.0±0.0 | 83.79±3.7 | 0.0±0.0 | 86.61±5.7 | 0.0±0.0 | 30.63±0.68 | 0.0±0.0 |
| Manipulated | 0.1 | 100.0±0.0 | 18.72±9.49 | 100.0±0.0 | 10.81±6.52 | 100.0±0.0 | 17.44±11.53 | 36.75±0.84 | 16.53±4.07 |
| Manipulated | 1.0 | 99.91±0.04 | 0.01±0.01 | 99.89±0.06 | 0.02±0.01 | 99.87±0.07 | 0.02±0.01 | 33.22±1.15 | 1.19±0.73 |

Table 5: Reproduced results of the Sequence-to-Sequence tasks with their respective error margins over the different seeds.

model even though their attention scores are low. The embedding model is the one with the least amount of trainable parameters. Masking a relatively high number of these parameters could be the cause for the dip in accuracy.

is lowthrough recurrent feedback connections. Because there are no recurrent connections in the simple embedding model, it does not manage to maintain good performance when attention weights are decreased for the sst-wiki task.

The results from the sequence-to-sequence tasks also correspond very well with the original paper. This is another support to the claims of the original paper that the attention mechanism can be manipulated in sequence to sequence tasks and that that process can force information through different *channels* within the model.

## 5.2 Reproducability

The overall reproducability of the paper was in our minds adequate. The code was, in the cases that it was available, easy to set up, easy to run and well documented. It has to be noted, though, that the transformer code was not available from the beginning. After some correspondence with the original authors, this issue was quickly resolved. However, proper functions to anonymize have yet to be added to the code. Secondly, a portion of the data that the authors used were private Graduate School Reference Letters and therefore not accessible. Using private data makes it practically impossible to reproduce that part of the paper. Lastly, the human study seemed unnecessary and its results unfounded, as it only concerned the answers of three people. It added bulk to an unstructured paper, which in our minds, would have benefited from less individual projects.

## 6 Conclusion

In conclusion, the results of the reproduction do support the results of Pruthi et al. [8] in almost all cases. This would mean that they uphold the original claims as well, were it not that we have some doubts whether the models are in fact learning to deceive.

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
