# OpenReview forum: "Learning to Deceive with Attention-Based Explanations"
_ML_Reproducibility_Challenge/2020 — Reject_

### Official Review · AnonReviewer3 · 2021-02-26
**Review for ML Reproducibility Challenge: Learning to Deceive with Attention-Based Explanations**

**Rating:** 4
**Confidence:** 3

**Review:**

This paper discusses reproducing the work included in Learning to Deceive with Attention-Based Explanations by Pruthi et al. Pruthi et al. claim that attention weights can easily be manipulated without significancy accuracy loss and that human subjects can be deceived by this attention weights. The paper focuses on reproducing the first part of the claim as the second part requires a user study.

The paper focuses on three models: embedding, biLSTM, and BERT. There are two evaluation tasks: classification (occupation, pronoun-based, sentiment analysis), and sequence-to-sequence. The paper shows that the results do reproduce the results fairly well, and the reproducibility aspect was acceptable (with quick responses from the authors).

Overall, the writing style in this paper needs to improve. There are many grammatical mistakes, and the paper does not flow very well. The first half of the paper did a good job of explaining the problem and the models used. However, the second half of the paper, including the results and reproducibility portions, needs more work. I would have like to see the differences in training time / evaluation time, better explanation of Table 4 (especially what the column names are), and more information on what correspondence occurred with the original authors (did they give a more thorough code? define the anonymization functions better?)

**Familiar With The Original Paper:**

I have not read the original paper

**Reproducibility Summary:**

Report has summary

---

### Official Review · AnonReviewer1 · 2021-02-27
**Good effort in reproducing the results, the content needs to be improved**

**Rating:** 4
**Confidence:** 4

**Review:**

The report describes the efforts in replicating the results of "Learning to Deceive with Attention-Based Explanations", where it is shown how 1) attention weights can be manipulated without loss in performance and 2) humans can be deceived by the obtained attention weights. The report describes the efforts in replicating the first claim. The authors of the report used the code of the original paper to replicate the results, adding the transformer part and missing anonymization functions. Overall, the results have been extensively replicated and the experiments well described. There are however some parts not well written (e.g. typos, half-sentences, missing links),  with missing explanations (e.g. multi-class sentiment analysis, transformer, anonymization) and with critiques to the claims of the paper (e.g. importance of the user study, capabilities of deceiving of the attention weights) not supported by experiments. Overall, I think the reproduction effort has been well conducted, but the report needs to be improved (see weaknesses below) to be accepteded. Below I detail what I think are the strengths and the weaknesses of the report.

Strengths:
+ The authors managed to replicate all the quantitative results of the main paper (on public datasets) with fair efforts for the missing components they added.
+ The experiments are very detailed, from the data splits to the hyperparameters used.
+ Discussions on the difference between the reproduced and the original results are also thorough and well justified.

Weaknesses:
- Maybe it is due to the lack of time, but the report misses careful proofreading. There are Tables not well linked (e.g. lines 136,172) wrongly cited equations (Eq. 0, line 74), typos (e.g. "Reproducability", line 207), missing end of sentence points (line 182), upper case start of sentences (e.g. line 161), half-sentences (line 202). Proofreading the report is necessary to ensure its quality.

- In Table 3 multiple numbers have an empty standard deviation for the A.M. column. Why is this the case? The table looks a bit weird with all the empty space after some +-, thus it would be good to add those values (preferably) or eliminate the +-.

- Some parts and definitions are not clear. For instance, I was not able to find definitions for A.M. (Table 3-4) and I (equation "0"). Similarly, the report mentions an extension of the paper by performing multi-class sentiment analysis (line 89) that however is not detailed in the following section and is not reported in the results (since the comparison is with the values reported in the original paper). These details should be included to avoid misconceptions.

- The attention weights \alpha are computed from the dot product between QK^T  "softmaxed". However, line 54 reports the non-softmaxed version of the dot-product to compute the attention, which is inaccurate. I would suggest the authors
to re-define the attention A by explicitly showing the contribution of \alpha and how it is computed.

- The difficulty of the reproduction due to 1) missing transformer code and 2) missing anonymization (lines 209-210) are not extensively described in the report. Since the report should highlight the encountered difficulties (if any) in reproducing the original results, I would extend section 2 to include a discussion of any components the authors needed to add to reproduce the results + every effort (e.g. missing libraries, dataset set up) that was required to reproduce them.

- The scope of the report is to reproduce the quantitative results of the original work, without reproducing the user studies. While I agree that the three subjects of the original paper do not constitute a large set allowing drawing general conclusions, I do not think it is fair to say that 1) the human study is unnecessary and unfounded (lines 213-215) and 2) raising doubts on the capabilities to deceive of the model (line 218). These are personal thoughts with no scientific/experimental grounds, since not results are shown to support that humans might not be deceived. I strongly suggest removing any claim not supported by 1) experiments 2) experience in the reproduction.

- The sentence in lines 214-215: "it added bulk to an unstructured paper, which in our minds, would have benefited from less individual projects"  is non-sensical and not founded anyway since 1) the unstructured paper is the original one? (peer-reviewed and accepted to ACL 2020) 2) what are the individual projects?

**Familiar With The Original Paper:**

I have not read the original paper

**Reproducibility Summary:**

Report has summary

---

### Official Review · AnonReviewer2 · 2021-03-01
**A good work**

**Rating:** 7
**Confidence:** 3

**Review:**

The authors did a good work, they joggle round missing data and came up with same result as the original authors.

**Familiar With The Original Paper:**

I have not read the original paper

**Reproducibility Summary:**

Report has summary

---

### Decision · Program_Chairs · 2021-03-31

**Decision:**

Reject

**Comment:**

Overall reviews and/or the paper content not good enough for the AC to recommend to the journal.